# Associations of Social Networks with Physical Activity Enjoyment among Older Adults: Walkability as a Modifier through a STROBE-Compliant Analysis

**DOI:** 10.3390/ijerph20043341

**Published:** 2023-02-14

**Authors:** Nestor Asiamah, Simon Mawulorm Agyemang, Cosmos Yarfi, Reginald Arthur-Mensah Jnr, Faith Muhonja, Hafiz T. A. Khan, Kyriakos Kouveliotis, Sarra Sghaier

**Affiliations:** 1Division of Interdisciplinary Research and Practice, School of Health and Social Care, Colchester CO4 3SQ, UK; 2Department of Gerontology and Geriatrics, Africa Centre for Epidemiology, Accra North P.O. Box AN 18462, Ghana; 3Department of Science/Health, Physical Education and Sports, Abetifi Presbyterian College of Education, Abetifi-Kwahu P.O. Box 19, Ghana; 4Department of Physiotherapy and Rehabilitation Sciences, University of Health and Allied Sciences, Ho PMB 31, Ghana; 5Department of Nursing and Midwifery, Faculty of Health and Allied Sciences, Pentecost University, Accra P.O. Box KN 1739, Ghana; 6Department of Community Health, School of Public Health, Amref International University, Nairobi P.O. Box 27691-00506, Kenya; 7College of Nursing, Midwifery, and Healthcare, University of West London, Paragon House, Boston Manor Road, Brentford TW8 9GB, UK; 8Berlin School of Business and Innovation, Academic Affairs, 97-99 Karl Marx Strasse, 12043 Berlin, Germany

**Keywords:** physical activity, active social networks, sedentary social networks, physical activity enjoyment, older adults, Ghana

## Abstract

The available evidence suggests that social networks can contribute to physical activity (PA) enjoyment, which is necessary for the maintenance of PA over the life course. This study assessed the associations of active and sedentary social networks with PA enjoyment and ascertained whether walkability moderates or modifies these associations. A cross-sectional design compliant with STROBE (Strengthening the Reporting of Observational Studies in Epidemiology) was employed. The participants were 996 community-dwelling older Ghanaians aged 50 years or older. A hierarchical linear regression analysis was used to analyse the data. After adjusting for age and income, the study found that the active social network size (β = 0.09; *p* < 0.05) and sedentary social network size (β = 0.17; *p* < 0.001) were positively associated with PA enjoyment. These associations were strengthened by walkability. It is concluded that active and sedentary social networks may better support PA enjoyment in more walkable neighbourhoods. Therefore, enabling older adults to retain social networks and live in more walkable neighbourhoods may be an effective way to improve their PA enjoyment.

## 1. Introduction

Physical activity (PA) has been evidenced to protect the individual from long-term conditions such as hypertension and diabetes [1,2,3]. A trajectory of PA can support optimal health across the lifespan [4,5], so the maintenance of PA is a hallmark for ageing in good health. Physical inactivity (PI), defined as the non-achievement of recommended PA levels [1,6], can increase the risks of early mortality and the above chronic conditions. A generally recommended level of PA for older adults is 75–150 min of vigorous-intensity PA or 150–300 min of moderate-intensity PA per week [7]. For older people to maximize the benefit of PA, they must meet this or a relevant recommended level of PA and avoid sitting too much. A change in life goals, frailty, functional limitations, and low PA enjoyment are among the most pronounced determinants of PI in older populations [8,9,10].

PA enjoyment is a feeling of joy, pleasure, or fun in PA [11]. It is a sustained emotion associated with the habit of exercise or a trajectory of PA [12,13], which means that people with high PA enjoyment would maintain the habit of exercising over the course of life. A growing body of research [13,14,15,16] has shown that PA enjoyment is necessary for the maintenance of a trajectory of PA. Studies have also shown that many people fail to exercise and meet recommended PA levels because they do not enjoy PA [14,17]. Thus, PA enjoyment plays a central role in maintaining PA in the ageing process. An active lifestyle is necessary for ageing in optimal health [18,19], so healthy ageing interventions ought to help older adults to enjoy PA across their lifespan.

Some studies [14,15,20] indicate that social networks play a role in PA enjoyment. In a review undertaken in the United States (US), for example, adolescents reported the relevance of PA enjoyment to PA [15]. In another study undertaken in the United Kingdom (UK), older adults reported experiences suggesting that PA enjoyment is necessary for maintaining PA [14]. Though the above studies imply or suggest that social networks can influence PA enjoyment, no study has assessed the association between social networks and PA enjoyment among older adults. It has been argued that sedentary and active social networks affect PA and its enjoyment in different ways, with the latter believed to better support PA enjoyment [14,15].

Active social networks are social ties (e.g., friends, blood relations, neighbours, workmates, or acquaintances) who encourage or support their friends to perform PA [8,20]. These social networks are not necessarily always active, but they regularly exercise and influence or support their social connections to avoid sedentary behaviour [8,20]. Sedentary social networks, compared with active social networks, are less active and compel or encourage others to perform sedentary behaviour [8]. These networks may be occasionally active but often perform long or successive episodes of sedentary behaviour [8,20]. As their sedentary behaviour may be due to personal factors (e.g., having a job that requires sitting, and living in a neighbourhood that does not support PA) [8], they can become active if their situations change and may occasionally support or encourage the PA of others. In view of these dynamics, it is unclear whether sedentary and active social networks can influence PA enjoyment in the same way; hence, a study assessing the associations of these networks with PA enjoyment was necessary. The first objective of this study was, therefore, to examine the above associations for the first time.

As mentioned above, the neighbourhood environment can influence the sedentariness of social networks and how long one remains inactive. Person–environment (P-E) fit models also recognise the pivotal role of the neighbourhood in the effort of individuals to maintain PA [21,22]. PA performed in the neighbourhood (i.e., neighbourhood-level PA) can be more enjoyable compared to PA performed indoors [13,23] since the neighbourhood offers aesthetic attributes (e.g., parks, lawns, and neighbourhood architecture). It is, thus, likely that social networks may better support PA enjoyment in more walkable neighbourhoods characterised by aesthetics, sidewalks, services, and psychosocial factors such as safety, trust, and reciprocity [8]. Walkability encompasses high residential density, street connectivity, and mixed land use (i.e., commercial and domestic uses) [24]. As both walkability and social networks can influence PA enjoyment, they can interact to influence older adults’ enjoyment of PA. This interaction forms the basis of the potential moderation role of walkability in the association of active and sedentary social networks with PA enjoyment. Though it has implications for ageing, this moderating role has not been examined. The second objective of this study, therefore, was to assess this potential moderation.

Thus, the purpose of this study was to assess the associations of social network types (i.e., sedentary social networks and active social networks) with PA enjoyment and ascertain whether these associations are moderated by walkability. The study’s research questions were (1) is there an association between active social networks and PA enjoyment? (2) is there an association between sedentary social networks and PA enjoyment? (3) is the relationship between active social networks and PA enjoyment moderated by walkability? and (4) is the relationship between sedentary social networks and PA enjoyment moderated by walkability? To maximise the significance of this study and better guide future gerontological research, we employed a cross-sectional design compliant with the STROBE (i.e., Strengthening the Reporting of Observational Studies in Epidemiology) checklist. Implications of the above relationships for healthy ageing are delineated.

## 2. Methods and Materials

### 2.1. Design

A STROBE-compliant cross-sectional design was adopted in this study. This design included a hierarchical linear regression (HLR) analysis for testing hypotheses and performing sensitivity analyses. Figure 1 is a flowchart of our design.

### 2.2. Study Population, Sample, and Selection

The study population was older adults who were permanent residents in two communities (i.e., Abetifi and Ho) in Ghana. We had no sampling frame for this study; hence, we did not know the study’s population size. Previous research [8,25] was followed to estimate the minimum sample size necessary with relevant statistics (i.e., power = 0.8, significance = 0.05, and effect size = 0.2) and G*Power software. The sample size reached for HLR with a maximum of 11 predictors was 95. To maximise the power of our tests, we aimed to gather data on all individuals who met the inclusion criteria. A total of 1003 individuals met the inclusion criteria, which are (1) the ability to walk independently for at least 10 min (i.e., this study had to focus on those who could perform PA and rate their PA enjoyment); (2) having a minimum of a basic education qualification, which evidenced participants’ ability to complete questionnaires in English, (3) being aged 50 years or older, and (4) willingness to participate in the study voluntarily. We selected the 1003 eligible older adults at community centres through structured interviews conducted by two of the researchers (i.e., SMA and CY). The length of an interview ranged from 5 to 10 min.

### 2.3. Measurement and Operationalisation

Active social network size and sedentary social network size were measured as continuous variables following a previously used method [8]. As older adults could have memory limitations, they were asked to report social networks from the last week’s activities. Section A.1 shows the specific items (questions) used to measure these variables. Physical activity enjoyment was measured with the 18-item Physical Activity Enjoyment Scale with seven numeric rating levels (i.e., 1, 2, 3, 4, 5, 6, and 7), where higher levels denote higher PA enjoyment. This scale was adopted in whole from a previous study [12] and produced a satisfactory internal consistency (Cronbach’s alpha = 0.87) in the current study. Section A.2 shows the items of this scale. Walkability was measured with the 11-item Australian version of the Neighbourhood Environment Walkability Scale (NEWS-AU) with five descriptive anchors (i.e., 1—strongly disagree, 2—disagree, 3—somewhat agree, 4—agree, and 5—strongly agree). This tool was adopted in whole from a previous study [8] and produced satisfactory internal consistency (Cronbach’s alpha = 0.77). It was used in this study because it is short and produced satisfactory psychometric properties in a sample of older adults in Ghana [8,26]. Section A.3 shows items measuring walkability. Data on active social network size, walkability, and PA enjoyment were generated by adding up their respective items.

Other variables measured were age, gender, self-reported health, chronic disease status, context experience (i.e., how long participants had lived in their current neighbourhood), income, and marital or relationship status. These variables were measured as potential confounders because the literature [5,8,23] identify them as factors that could influence social network size and its relationship with PA. Gender (coded: male—1; female—2), chronic disease status (coded: none—1; one or more—2), relationship status (coded: not in a relationship—1; in a relationship—2), and self-reported health (coded: poor—1; good—2) were measured as dichotomous categorical variables. These variables were coded into dummy-type variables to support HLR analysis. Age (in years) was measured as a continuous variable by asking the participants to report their age. Income was measured as a continuous variable by asking the participants to report their gross individual monthly income in Ghana cedis. Education was measured by asking the participants to report their years of schooling. Finally, context experience was measured following previous research [26] by asking the participants to report how long (in years) they had lived in their current neighbourhood.

### 2.4. The Questionnaire and Measures against Common Method Bias

We collected data for this study with a self-reported questionnaire comprising three main sections. The first section captured the research aim, instructions for completing the survey, and the study’s ethical statement. The second section presented questions measuring the personal characteristics or confounding variables, whereas the third section captured measures of active social networks, sedentary social networks, PA enjoyment, and walkability. Common method bias (CMB) is one of the major threats to the internal validity of cross-sectional studies as it can result in associations that are due only to response bias [26,27]. Our first step against CMB was at the study design stage where the questionnaire was structured in harmony with recommendations [26]. Each section of the questionnaire was presented as a unique part separated from other sections. Our second step against CMB was the one-factor method [26,27], a statistical procedure in which exploratory factor analysis with varimax rotation was used to explore the factor structures of the psychometric scales. This method evidences the absence of CMB if a factor solution of two or more factors is produced, and the first factor accounts for a variance of less than 40% [26]. This procedure produced a satisfactory factor solution for the two scales: walkability (4 factors, total variance = 60.3%, variance accounted by the first factor <30%, factor loading ≥0.5) and PA enjoyment (6 factors, total variance = 62.7%, variance accounted by the first factor <30%, factor loading ≥0.5).

### 2.5. Data Collection and Ethics

This study received an ethics review and clearance from an institutional ethics review board in Accra (# 002-10-2022-ACE). All the participants provided written informed consent to participate in this study. Data collection in each community was coordinated by two of the researchers (CY and SMA) and their research assistants. Questionnaires were hand-delivered to the participants who completed the surveys and handed them back to the research assistants. The questionnaires were administered over four weeks (12 November to 10 December, 2022). Out of 1003 questionnaires returned, 7 of them were discarded because they were completed halfway. Thus, 996 questionnaires were analysed.

### 2.6. Statistical Analysis Methods

Data were analysed in two phases with SPSS software (IBM SPSS Inc., New York, NY, USA, version 28). We employed HLR analysis in this study because this type of multiple linear regression is the ultimate tool for performing our two sensitivity analyses [28]. It also enabled us to fit regression models involving multiple predictors in a stepwise way. In the first phase of the analysis, we summarised the data with descriptive statistics (i.e., means and frequencies), assessed relevant assumptions for using HLR analysis, and performed the first sensitivity analysis to screen for the ultimate confounding variables. The specific assumptions assessed and met were the normal distribution of the data, linearity of the hypothesised relationships, independence of errors, multicollinearity, and homogeneity of variances around the regression line [26,28]. Section B.1 shows the steps followed to assess and meet all assumptions. The data were analysed without removing the missing items because less than 4% of the data were missing [28]. The first sensitivity analysis, which was adopted from previous research [28], was aimed at screening the potential variables for the ultimate confounders (i.e., variables more likely to confound the hypothesised associations). This approach is more robust and useful in a situation where multiple potential confounders are considered. Section B.2 shows the individual steps taken in this analysis. Age and income were identified in this analysis as the ultimate confounders and were, therefore, infused in the final analysis.

In the second phase, the hypotheses depicted in Figure 2 were tested. Firstly, Pearson’s bivariate correlation coefficients between relevant variables were generated as a basis of the HLR. Two categories of models were then fitted; the first category comprised four non-adjusted models (i.e., Models 1–4) excluding the ultimate confounding variables, whereas the second category comprised four adjusted (ultimate) models (i.e., Models 5–8) that infused the ultimate confounders. Models 1 and 2 tested the first (H1) and second hypotheses (H2), whereas Models 3 and 4 assessed the third (H3) and fourth hypotheses (H4), respectively. Models 5–8, on which this study’s conclusions are based, were built upon Models 1–4 by incorporating the ultimate confounders. Following a previous study [28], we performed a second sensitivity analysis by comparing the regression weights of the adjusted and non-adjusted models to see the potential influence of the confounders on the adjusted model.

To assess moderation (i.e., H3 and H4), we followed previous research [26] to compute two interaction terms (i.e., ASNSxNW and SSNSxNW) using the ‘compute variable’ function in SPSS. ASNSxNW was the interaction between active social network size and walkability, whereas SSNSxNW was the interaction between sedentary social network size and walkability. Within Models 2, 3, 6, and 8, we tested the moderating roles of interest by assessing the association between these interaction terms and PA enjoyment. As performed in a previous study [26], we assessed a ‘pure moderation’ as we were only interested in whether walkability can change the regression weight between each of the predictors (i.e., active social network size and sedentary social network size) and PA enjoyment. The statistical significance of all tests was detected at a minimum of *p* < 0.05.

## 3. Findings

Table 1 shows a summary of the data on all the variables. About 50% of the participants were females, and the average age was about 66 years (mean = 66.34; SD = 10.51). The average PA enjoyment was 77 (mean = 77.21; SD = 19.4). The summary statistics on other variables can be seen in Table 1. Table 2 shows Pearson’s correlations among PA enjoyment, active social network size, sedentary social network size, and the ultimate confounders (i.e., income and age). There was a positive but weak correlation between PA enjoyment and active social network size (r = 0.195; *p* < 0.001; two-tailed) as well as the sedentary social network size (r = 0.185; *p* < 0.001; two-tailed). Thus, higher PA enjoyment was associated with a higher active social network size and sedentary social network size. There was a moderate positive correlation between PA enjoyment and walkability (r = 0.377; *p* < 0.001; two-tailed), which suggests that PA enjoyment was associated with higher walkability.

Table 3 shows the regression results. The first ultimate model (i.e., Model 5) shows a positive association between active social network size and PA enjoyment (β = 0.094; t = 2.79; *p* < 0.05) after adjusting for income and age, which confirms that PA enjoyment was higher with the active social network size. In Model 6, the sedentary social network size was positively associated with PA enjoyment (β = 0.167; t = 5.49; *p* < 0.001), which connotes that PA enjoyment is higher at a higher sedentary social network size. In Model 7, the interaction term ASNSxNW was positively associated with PA enjoyment (*p* < 0.001); the standardised regression weight in Model 5 increased from 0.094 to 0.137, which represents a 78% increase in the effect size due to walkability. This result implies that walkability enhanced the strength of the association between the active social network size and PA enjoyment by about 78%. In Model 8, the interaction term SSNSxNW was positively associated with PA enjoyment (*p* < 0.001); the standardised regression weight increased from 0.094 to 0.189 in Model 8, which represents a 101% change in the effect size due to walkability. Thus, walkability strengthened the association between the sedentary social network size and PA enjoyment by about 101%.

## 4. Discussion

### 4.1. Discussion of Findings

This study evaluated the associations of active and sedentary social network size with PA enjoyment and ascertained whether walkability can modify these associations among older adults. These relationships were tested with a STROBE-compliant design including relevant sensitivity analyses.

This study found a positive association between active social network size and PA enjoyment, suggesting that PA enjoyment was higher among older adults with more active social networks. This result confirms the first hypothesis (H1) and implies that having more active social networks may be associated with a higher enjoyment of PA. Supporting this evidence are some studies that have explored the roles of social networks in social and physical activities [13,15,16,29]. A qualitative study in Brazil reports the experiences of older adults suggesting that active friends and neighbours contributed to PA enjoyment [29]. Other studies on older adults [30,31] indicate that PA is more enjoyable and sustainable if performed with social network members. A literature review showed that social networks contributed to the enjoyment of PA, though these networks comprised younger adults and may have included both active and sedentary networks. Deductively, the positive association between active social networks and PA enjoyment may not be limited to older adults.

A positive association between sedentary social network size and PA enjoyment was also confirmed, which connotes that PA enjoyment was higher among older adults with more sedentary social networks. This result confirms the second hypothesis (H2) and is supported by some researchers who reasoned, based on their empirical results, that sedentary social networks can encourage PA remotely through messaging and phone calls [8,32]. To explain, sedentary social networks, who know about the benefit of PA but do not exercise for reasons beyond their control (e.g., being frail or busy) [8], can remind their peers of the importance of keeping active and encourage them to exercise regularly. It has been argued that knowledge about the health benefits of exercise and one’s involvement in PA based on this knowledge can make PA enjoyable [33]. As our results indicate, older adults had both active and sedentary social networks, so PA enjoyment attributable to active social networks can also overlap with sedentary social networks. As mentioned earlier, sedentary social networks may also be active depending on their context and situation (e.g., changing jobs that involved too much sitting and alternating their residence between neighbourhoods of low and high walkability), so it is possible that these networks contributed to PA and its enjoyment.

Walkability strengthened the associations between the active and sedentary social network size and PA enjoyment, which means that these social networks better contributed to PA enjoyment in more walkable neighbourhoods. This result supports the third and fourth hypotheses (i.e., H3 and H4) and stems from a positive association between walkability and social networks, which has been confirmed empirically [8,34]. A study conducted in Ghana [34] suggests that people in more walkable neighbourhoods may have more social ties who encourage and contribute to PA. As both active and sedentary social networks contribute to PA enjoyment, the association of these networks with higher walkability can be expected to be stronger. The above thoughts are congruent with the viewpoint that PA performed outdoors or around green space is more enjoyable compared with PA performed indoors [33]. P-E fit models, such as the life–space concept [22] and the Context Dynamics in Ageing [35] framework, assume that the neighbourhood offers attractive or aesthetic features that would make PA performed within it more appealing, sustainable, and enjoyable. This theoretical argument is supported by our data.

Our results have implications for practice and research. This study reinforces the importance of interventions and investments aimed at improving walkability, especially its aesthetic properties known to ease PA and facilitate PA enjoyment [8,33]. The worth of these investments is justified by the potential influence of walkability on PA enjoyment through social networks. Enabling ageing people to preserve their social networks is imperative as older adults are more likely to avoid PA if they cannot enjoy it. Moreover, the possibility of older adults not enjoying PA is higher owing to their physiological limitations and frailty. Therefore, interventions enabling them to enjoy and maintain PA into later life may be necessary. Worth noting is the relative influences of active and sedentary social networks on PA enjoyment. Though active social networks were more strongly associated with PA in the non-adjusted model (see Table 3), sedentary social networks accounted for a larger regression weight on PA enjoyment after adjusting for the ultimate covariates. Thus, more than 106% of the influence of active social networks on PA enjoyment in the non-adjusted model was due to age and income. A key lesson is that the contribution of active social networks to PA enjoyment may depend largely on age, income, and possibly other personal factors not considered in this study. As such, future researchers are encouraged to control for these personal factors in testing the associations of active and sedentary social networks with PA enjoyment.

### 4.2. Strengths and Limitations

This study was a cross-sectional study, which means it could not establish causation between the variables [36]. Even so, it provides findings that could inform policy and practice. Its statistics (e.g., regression weights) may also be used in calculating the necessary sample size for future studies. The sampling method adopted was non-probabilistic, so the sample may not be representative of older adults in Ghana. The sample is also relatively small compared to studies utilising national and regional samples. Future studies are, therefore, encouraged to utilise larger and more representative samples. Subjective measures were utilised in this study, which means bias due to poor recall of past events was probable. We, nevertheless, tried to avoid this with our steps against CMB and by asking older adults to report recent or current experiences and events (e.g., activities performed in the last 7 days). We admit that the other factors (e.g., the individual’s PA and sedentary behaviour) we did not measure may confound our results, so we call for future studies adjusting for these factors. Despite these limitations, this study has several strengths that make it outstanding.

Firstly, this was the first study to examine the associations of active and sedentary social networks with PA enjoyment; there has been no empirical assessment of whether these two categories of social networks can be associated with PA enjoyment. This study was STROBE-compliant, which means it followed all reporting guidelines for cross-sectional studies. As most studies did not follow this checklist [28], this study is a model for future cross-sectional studies. Appendix C shows the specific recommendations of STROBE which were met.

## 5. Conclusions

After adjusting for the ultimate confounders (i.e., income and age), both the active network size and sedentary social network size were positively associated with PA enjoyment, which means that PA enjoyment was higher with higher active and sedentary social networks. We, therefore, conclude that having more active and sedentary social networks may be associated with PA enjoyment. Walkability strengthened the association of active and sedentary social networks with PA enjoyment. Thus, active and sedentary social networks may more significantly support PA enjoyment in more walkable neighbourhoods. Future studies incorporating the confounding variables we could not measure are needed.

## Figures and Tables

**Figure 1 ijerph-20-03341-f001:**
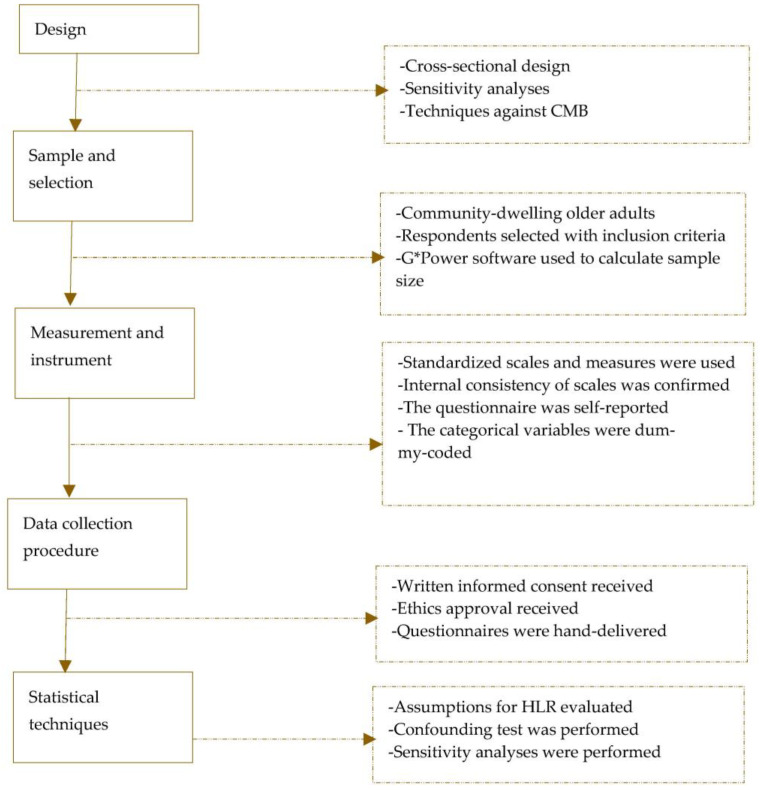
Elements of the STROBE-based design. HLR—hierarchical linear regression; CMB—common method bias.

**Figure 2 ijerph-20-03341-f002:**
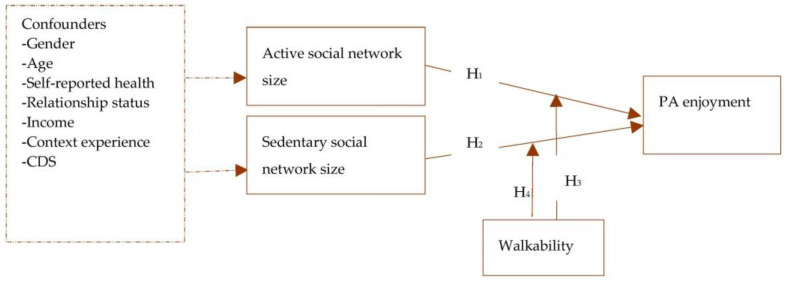
The association between active social networks, sedentary social networks, walkability, and PA enjoyment **Note**: Broken arrows represent potential confounding; CDS—chronic disease status; PA—physical activity; H_1_—active social network size is associated with PA enjoyment; H_2_—sedentary social network size is associated with PA enjoyment; H_3_—walkability moderates the association between active social network size and PA enjoyment; and H_4_—walkability moderates the association between sedentary social network size and PA enjoyment.

**Table 1 ijerph-20-03341-t001:** A summary of the data with descriptive statistics (n = 996).

Variable	Group	n/Mean	%/SD
Categorical variables			
Gender	Male	495	49.7
	Female	501	50.3
	Total	996	100
Self-reported health	Poor	337	33.84
	Good	650	65.26
	Missing	9	0.9
	Total	996	100
Relationship status	No	245	24.6
	Yes	751	75.4
	Total	996	100
Chronic disease status	None	353	35.44
	One or more	638	64.06
	Missing	5	0.5
	Total	996	100
Continuous variables			
Income (GHS)	---	787.14	933.37
Age (yrs)	---	66.34	10.51
Context experience (yrs)	---	34.21	24.81
Active social network size	---	4.04	3.96
Sedentary social network size	---	1.04	1.31
Walkability	---	36.11	5.09
Physical activity enjoyment	---	77.21	19.4
Education (yrs)	---	12.09	3.90

Note: --- not applicable; n—frequency; SD—standard deviation; the mean and SD apply to only continuous variables, whereas the frequency and % apply to only categorical variables.

**Table 2 ijerph-20-03341-t002:** Pearson’s correlations between active social network size, sedentary social network size, PA enjoyment, walkability, and the ultimate confounders (n = 996).

Variable	1	2	3	4	5	6
1. Physical activity enjoyment	1	0.195 **	0.185 **	0.377 **	0.298 **	−0.130 **
2. Active social network size		1	0.417 **	0.098 **	0.368 **	−0.349 **
3. Sedentary social network size			1	0.157 **	0.056	−0.181 **
4. Walkability				1	0.055	0.065 *
5. Income (GHS)					1	−0.313 **
6. Age (years)						1

** *p* < 0.001; * *p* < 0.05.

**Table 3 ijerph-20-03341-t003:** The associations between active social network size, sedentary social network size, physical activity enjoyment, and walkability (n = 996).

Model	Predictor	Regression Weights			95% CI	Model Fit			
		B	SE	β(t)		R^2^	Adjusted R^2^	Durbin–Watson	F-Test
1	(Constant)	73.367	0.862	(85.09) **	±3.384	0.038	0.037		38.96 **
	Active social network size	0.952	0.153	0.194(6.24) **	±0.598	---	---	---	---
2	(Constant)	74.381	0.772	(96.40) **	±3.029	0.034	0.033		34.89 **
	Sedentary social network size	2.72	0.461	0.184(5.91) **	±1.807	---	---	---	---
3	(Constant)	72.829	0.841	(86.57) **	±3.302	0.052	0.051		55.00 **
	ASNSxNW	0.03	0.004	0.229(7.42) **	±0.015	---	---	---	---
4	(Constant)	74.013	0.761	(97.29) **	±2.985	0.045	0.044		47.02 **
	SSNSxNW	0.083	0.012	0.213(6.86) **	±0.047	---	---	---	---
5	(Constant)	72.948	4.408	(16.39) **	±17.301	0.098	0.095	1.72	35.76 **
	Active social network size	0.46	0.165	0.094(2.79) *	±0.646	---	---	---	---
	Income (GHS)	0.005	0.001	0.259(7.82) **	±0.003	---	---	---	---
	Age (yrs)	−0.028	0.061	−0.015(-0.46)	±0.238	---	---	---	---
6	(Constant)	71.113	4.229	(16.82) **	±16.597	0.117	0.115	1.83	43.97 **
	Sedentary social network size	2.461	0.448	0.167(5.49) **	±1.758	---	---	---	---
	Income (GHS)	0.006	0.001	0.286(9.12) **	±0.002	---	---	---	---
	Age (yrs)	−0.017	0.059	−0.009(-0.29)	±0.231	---	---	---	---
7	(Constant)	71.316	4.351	(16.39) **	±17.075	0.106	0.103	1.92	39.17 **
	ASNSxNW	0.018	0.004	0.137(4.13) **	±0.017	---	---	---	---
	Income (GHS)	0.005	0.001	0.247(7.47) **	±0.002	---	---	---	---
	Age (yrs)	−0.012	0.06	−0.006(-0.19)	±0.236	---	---	---	---
8	(Constant)	70.489	4.201	(16.78) **	±16.488	0.125	0.122	1.81	47.29 **
	SSNSxNW	0.074	0.012	0.189(6.26) **	±0.046	---	---	---	---
	Income (GHS)	0.006	0.001	0.282(9.01) **	±0.002	---	---	---	---
	Age (yrs)	−0.011	0.059	−0.006(-0.189)	±0.23	---	---	---	---

** *p* < 0.001; * *p* < 0.05; NW—neighbourhood walkability; ASNS—active social network size; SSNS—sedentary social network size; SE—standard error (of B); CI—confidence interval (of B).

## Data Availability

Our data will be made available by the corresponding author upon request.

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
