# Peer review of "Associations of Social Networks with Physical Activity Enjoyment among Older Adults: Walkability as a Modifier through a STROBE-Compliant Analysis"

_ijerph, 2023, doi:10.3390/ijerph20043341_

Round 1

Reviewer 1 Report

Introduction does not follow a clear and sequential structure, with several repetition in come concepts.

The scientific rationale behind the study has not been clearly derived. 

Clear purpose has not been declared. Hypotheses are missing. 

Spell out abbreviations first time they are mentioned, as HLR CMB.

Methodology has been well described.

How final sample has been included in analysis is not clear.

Figure 3 and 4 are very unclear.

Reference style has not been followed.

Finally the overall quality of scientific and academic writing style is very low.

Author Response

Please our comments are attached. 

Reviewer 2 Report

The article explores the relationship between the nature of the social network and the ability to walk among older Adults.

The authors may consider the following comments:

In line 60 it would be advisable to add " per week".

Few sentences, e.g. lines 90-94, are written in a difficult way to understand. It should be divided into shorter ones to make it more comprehensible to read.

In Figure 1, the boxes on the right are incomplete, the reader is not able to see the content. For example “hand- delivered?”

One of the caveats is age. In line 171, it says that people aged 50 and older were included in the study. It is not clear if both the title and further content in the group should be defined as elderly.

Try to avoid repetitions, like in line 234.

In results, for the section of strengths and limitations, the authors repeat some of the assumptions described in the methods. it should be reformulated and shortened.

Summary: The work is prepared substantively and methodologically in a correct way. My objections relate to the fact that the work has a relatively small impact on the current scope of knowledge in the subject matter. The obtained results are not revealing, but rather predictable. In addition, I would like to pay attention to the style in which the above work is written. The texts are difficult to read, complicated and the descriptions of the research steps performed are very long-winded. In the case of procedures described in previously published articles, some elements can be mentioned by adding appropriate citations. I would recommend a minor revision of the text to make it more readable.

Author Response

Our comments are attached please

Reviewer 3 Report

Thank you for receiving the invitation from the editor-in-chief to review, and the author needs to correct some content to maintain a certain level.

Abstract

1. Academic research recommends not to use “Anecdotal”.

2. “Sedentary social network size with PA enjoyment by 101%.” This is problematic, there is a negative correlation between sedentary and PA.

Introduce

1. The author failed to define the social network and lead to the primary purpose of this study, which is actually very vague.

2. Line88-96 cannot be connected with the previous text and must be revised.

3. Why did the author use walking?

4. The difference between active and sedentary social networks is not clear? It is possible that sedentary social networks can also be active (social expansive).

Method

1. Figure 1 is meaningless, it is just a flow chart, I hope the author redesigns the research structure.

2. HLR, CMB should explain the reason you use in the statistical method.

3. “Verbal informed consent” This is an unreal form of consent.

Result

1. Please use male and female for the full text.

2. Unfortunately, I still struggle with the novelty of this study/why it needed to be conducted. I also struggle with the fact that there are many possible confounding variables that could affect results, and that you did not measurements or differences in pre- and post tests for participants walking.

3. In the argumentation in Table 3, there are many deficiencies and root doubts. The reason lies in the lack of clear definition of social network, actually "sedentary social network" is positively correlated with physical activity?

Results & Discussion

-  As mentioned previously, there are many factors that were not measured and taken into consideration in analyses that could affect results.

-  Further, you need to talk more about how other external factors that could influence results were not addressed.

-  I do not understand how much authenticity can the questionnaire show, and whether to add more in-depth experimental research? What is novel about this?

References

Please use the correct format

Is an appendix required? Important content should be presented in the text.

Author Response

Please we have attached our response

Reviewer 4 Report

Manuscript ID: ijerph-2169537

Journal: International Journal of Environmental Research and Public Health

Associations of Social Networks with Physical Activity Enjoyment among Older Adults: Walkability as a Modifier Through a STROBE-Compliant Analysis.

Thank you for the opportunity to review this manuscript. This paper describes your work exploring the potential influence of social networks on the enjoyment of physical activity and also attempts to link neighbourhood walkability to moderation of that influence.

The study design and methods are well described. I was however troubled by two assumptions that appeared to underpin the methodology. 

1. That those that enjoy physical activity are physically active. While the importance that of enjoyment that you highlight is absolutely acknowledged, there are still many who are active and do not enjoy the process - just the outcomes. I would have liked/expected some indicator of participants levels of PA - an IPAQ5 for example would provide a proxy indicator.

2. That walking was the PA of choice. Unless walking is my main activity I am not sure of the relevance of neighbourhood walkability. Again a simple question on preferred activities may have helped.

I recognise that you of course can not go back and redo your research, however, I would like to see you address these assumptions in your study rationalisation and/or limitations section. You have taken care with CMB after all.

L 381 - 383

You state that

‘This result confirms the first hypothesis (H1) and implies that active social networks contribute to the enjoyment of PA.’

Which is at odds with your acknowledgement in your limitations that you could not suggest causation between variables…..

L 406 - 408 

‘The above relationship could also be the result of some older adults enjoying PA due to their active peers while maintaining some sedentary social networks.’

I do not understand this sentence - perhaps a word or words are missing?

Conclusions

I am not really clear on your practical messages here. Your data and analysis does not identify participants with zero active or sedentary social network contacts and the influence that that might have on PA enjoyment. What I am saying is that you have provided a very detailed description of your statistical model and analyses but the external validity and practical implications are not clear. Perhaps you do not have enough detail to form those conclusions but I think this section needs revisiting. 

Minor point - Appendix A1 

Line 526

I think acquaintances rather than acquittances.

Author Response

Please refer to our response in the attachment 

Round 2

Reviewer 1 Report

Change heading 3. in Results 

Remove 4.1 sub headings

The manuscript has been improved based on previous report.

Author Response

Response to the Editor and Reviewers

We really appreciate the time spent by the editor and the review team on our manuscript; we are grateful to them for using their expertise to help us improve our manuscript. We have addressed the comments from the first reviewer as follows:

Change heading 3. in Results

Our response: Thank you for this. Heading 3 reads “Findings”, which is the way IJERPH presents this section. We guess you’re asking us to change it to “Results”, but this heading in our previous draft manuscript published in IJERPH was not accepted. So please allow us to stick to the journal’s standard, which is “Findings”.

Remove 4.1 sub headings

Our response: Yes, we agree that is a bit confusing. We’ve removed it.

The manuscript has been improved based on previous report.

Our response: Thank you for your time.